# Do Food Preservatives Affect Staphylococcal Enterotoxin C Production Equally?

**DOI:** 10.3390/ijms262311659

**Published:** 2025-12-02

**Authors:** Aleksandra Tabiś, Keun Seok Seo, Juyeun Lee, Joo Youn Park, Nogi Park, Jacek Bania

**Affiliations:** 1Department of Food Hygiene and Consumer Health Protection, Wrocław University of Environmental and Life Sciences, 50-375 Wrocław, Poland; jacek.bania@upwr.edu.pl; 2Department of Comparative Biomedical Sciences, College of Veterinary Medicine, Mississippi State University, Starkville, MS 39762, USA; seo@cvm.msstate.edu (K.S.S.); leej30@ccf.org (J.L.); jpark@cvm.msstate.edu (J.Y.P.); np509@msstate.edu (N.P.); 3Department of Cardiovascular and Metabolic Sciences, Lerner Research Institute, Cleveland Clinic, Cleveland, OH 44195, USA

**Keywords:** food preservatives, enterotoxin C, *Staphylococcus* spp.

## Abstract

Staphylococcal enterotoxins (SEs), particularly enterotoxin C (SEC), are potent superantigens primarily known for causing food poisoning, but recent studies have highlighted their potential role in immune-mediated intestinal diseases. Despite the widespread use of food preservatives, their influence on SEC production—especially from coagulase-negative staphylococci (CNS)—remains poorly understood. In this study, we evaluated the effects of commonly used preservatives, including sodium chloride, potassium nitrate, and sorbic acid, on the expression and production of SEC_3_ and SEC_epi_ in *Staphylococcus aureus* and *S. epidermidis*, respectively. Using ELISA and RT-qPCR, we analyzed toxin levels at both the protein and mRNA levels. Proliferation assays on human PBMCs assessed the mitogenic potential of culture supernatants. While sodium chloride and potassium nitrate did not significantly alter SEC levels or bacterial growth, only sorbic acid at 0.07% consistently inhibited both mRNA expression and protein production of SEC_3_ and SEC_epi_. Furthermore, supernatants from sorbic acid-treated cultures induced significantly lower PBMC proliferation. These results suggest that even sub-emetic concentrations of enterotoxins may have immunomodulatory effects, and sorbic acid could be a promising agent in mitigating such risks.

## 1. Introduction

*Staphylococcus aureus* is a known cause of staphylococcal food poisoning (SFP), which results from the ingestion of food containing staphylococcal enterotoxins (SEs). Despite the implementation of safety systems in the food industry, the reported number of intoxications remains high worldwide. In 2023, in Europe, 207 outbreaks, 2268 cases, and 113 hospitalizations due to foodborne illnesses caused by SEs were reported [1]. The Centers for Disease Control and Prevention (CDC) estimates that 241,148 cases of SFP occur each year in the U.S. [2]. According to EU regulation No. 1441/2007, the presence of SEs is monitored only in milk-derived products. Foods such as meat and meat products are not routinely tested for enterotoxin content. However, several studies have shown that the presence of staphylococcal toxins in these types of foods is not uncommon, and many SFP outbreaks have been attributed to the consumption of meat products containing SEs [1]. Following risk assessment, the primary challenge in mitigating the risk of *Staphylococcus aureus* intoxication is to first prevent the growth of the bacteria, and subsequently inhibit enterotoxin production, or at the very least, prevent the accumulation of enterotoxins in high concentrations (e.g., >10–20 ng) in food [3]. SEs are only slightly or not inactivated during food processing, storage, distribution, and preparation [4,5,6,7]. Standard processing of a significant number of meat products includes a curing procedure, during which the meat is treated with curing salt containing sodium chloride and nitrites. This procedure helps to preserve the meat from unwanted microflora and contributes to the development of the desired color in the final product. Food preservatives are concurrently recognized as stressors affecting the change in expression of virulence genes in bacteria [8]. Osmotic stress is typically induced using sodium chloride. Studies conducted on the expression of enterotoxins under the influence of NaCl pertain to *S. aureus* species [3,9,10]. Studies on the impact of curing salt on SE production have yielded ambiguous results, generally indicating that it is not highly effective in preventing SE production in meat [11,12]. Sorbic acid (SA) is also widely used in the food industry as a preservative, inhibiting the growth of yeast and mold [13,14]. The effect of SA was tested on the expression of selected enterotoxins only [3,13,15]. It was found that SA has a variable effect on SE production but does not significantly affect *S. aureus* growth [3].

Among the classical staphylococcal enterotoxins (SEA–SEE), enterotoxin A (SEA) is the most frequently detected in food products such as cheese, raw milk, seafood, fish, meat, and meat products [16,17]. However, among the enterotoxins produced by coagulase-negative staphylococci (CNS), the most commonly reported genes in the literature are sec, seh, sed, and seu [4,18]. Nevertheless, confirmed expression of the toxin in food has been demonstrated mainly for enterotoxin C (SEC), and only SEC has been fully characterized in terms of genomic location, stability, resistance to proteolytic digestion, and enterotoxigenic effect in in vivo studies [18,19,20]. The gene encoding for the *S. aureus* enterotoxin C orthologue (SEC_epi_) was found in certain strains of *S. epidermidis*, belonging to the coagulase-negative staphylococci group [21]. Despite significant differences in amino-acid composition and content of secondary structures between SEC_epi_ and *S. aureus* SEC variants, some SEC_epi_ properties such as superantigenic activity, and stability were found to be similar to its *S. aureus* orthologues [19,20]. The regulation of SEC_epi_ expression and the effect of external factors, including food preservatives on SEC_epi_ production remains unknown.

The aim of our research was to determine whether common food preservatives have a similar effect on the production of enterotoxin C produced by CNS as well as by coagulase-positive staphylococci. The expression of enterotoxin under the influence of sodium nitrate, sodium chloride, and sorbic acid was examined using *S. epidermidis* 4S strain isolated from food [22] and the reference strain *S. aureus* FRI 913.

## 2. Results

### 2.1. Growth and Determination of a Peak of Enterotoxin Production by S. epidermidis and S. aureus

The presence of 0.07% sorbic acid reduced the optical density of *S. epidermidis* 4S during the first 7 h (Figure 1A) (*p* < 0.001). No differences were observed after 24 h of *S. aureus* growth in the presence of food preservatives and a control. Dot blot analysis (Figure 1B) performed on post-culture fluids at individual time points showed that the optimal time for analyzing expression at the translational level of the toxins was the stationary phase, 24 h after the start of the culture. Due to the limited sensitivity of the dot blot method, further analyses were performed using the more sensitive ELISA method.

### 2.2. Effect of Food Preservatives on Secretion of SECepi and SEC_3_

SEC_epi_ protein levels produced by *S. epidermidis* were lower in the control CY medium and in CY containing 4.5% NaCl (*p* = 0.041), 4.5% potassium nitrite (*p* = 0.007), a mixture of 2% NaCl and 2% KNO_3_ (*p* = 0.009), and 0.07% sorbic acid (*p* = 0.0007), compared to SEC_3_ produced by *S. aureus* under the same conditions. It was found that production of SEC_epi_ and SEC_3_ by *S. epidermidis* and *S. aureus*, respectively, cultivated in CY containing NaCl and KNO_3_ at the concentrations used in the experiments was not changed compared to the control CY medium. In turn, *S. epidermidis* produced significantly less SEC_epi_ in medium containing 0.07% sorbic acid compared to the control medium (*p* < 0.001). The average concentration of SEC_epi_ secreted by *S. epidermidis* in 0.07% sorbic acid at 24 h was 6.8 ng/mL, while SEC_epi_ production in the control medium was 1850 ± 673 ng/mL. Also, in *S. aureus* 0.07% sorbic acid decreased the production of SEC_3_ protein to a concentration of 260.6 ng/mL compared to SEC_3_ production in the control medium, which accounted for 3621 ± 72 ng/mL (*p* < 0.001).

Bacterial numbers of both, *S. aureus* and *S. epidermidis* after 24 h were similar in control medium and media supplemented with the tested food additives (Figure 2A).

### 2.3. Effect of Food Preservatives on mRNA Expression of Secepi and sec_3_

To assess whether changes in enterotoxin production also occur at the transcript level, *sec* transcripts were determined 24 h after inoculation. The effect of the different treatments on sec mRNA production is presented in Figure 2B. At this time point, the data obtained were not as homogeneous as in the assessment of the amount of protein produced. The most significant reduction in the expression of enterotoxin C occurred after the addition of 0.07% sorbic acid to the medium. Both species showed decreased transcript levels of sec (*S. epidermidis p* = 0.0163 and *S. aureus p* = 0.0001).

*S. epidermidis* cultivated in CY + 4.5% NaCl broth showed a decrease in *sec_epi_* transcript levels (*p* = 0.0226). A significant increase in sec transcript levels was observed when 2% NaCl *(p* = 0.0002) and 4.5% NaCl (*p* = 0.0328) were added to the medium inoculated with *S. aureus*.

*S. epidermidis* appears to be more sensitive to the effects of nitrates; 2% potassium nitrate increased the *sec* transcript levels (*p* = 0.0042). An additive effect was observed in a mixture of 2% NaCl and 2% KNO_3_, with both species producing increased sec transcripts (*S. aureus p* = 0.091, *S. epidermidis p* < 0.0001). Comparing the two species, *S. aureus* produced significantly more enterotoxin C in the presence of 2% NaCl (*p* < 0.05) and 4.5% NaCl (*p* < 0.01). *S. epidermidis* responded to 4.5% KNO_3_ (*p* < 0.001) by increasing the level of the sec transcript compared to the coagulase-positive strain. A mixture of 2% NaCl and 2% KNO_3_ (*p* = 0.00045) also increased the level of the sec transcript compared to *S. aureus* FRI913.

### 2.4. Proliferation Assay

To assess the immunostimulatory effect of *S. epidermidis* and *S. aureus* SECs a PBMC proliferation assay using CFSE labeling was performed. Despite differences in interaction with different SEC orthologs and individual differences between PBMCs from different donors this method was chosen due to its high sensitivity in detecting subtle proliferative responses at the single-cell level. As demonstrated in Figure 3A, the CFSE-based assay revealed a measurable immunostimulatory effect at nanogram concentrations of enterotoxin C. This is particularly significant considering that emetic symptoms in humans are typically associated with the ingestion of several tens of nanograms of staphylococcal enterotoxins. Our findings indicate that, despite the substantial suppression of enterotoxin production by sorbic acid (Figure 2A), the residual levels of SEC were still sufficient to elicit a mitogenic response in PBMCs (Figure 3B,C), underscoring the potential biological relevance of even minimal toxin presence. As expected, culture supernatants from 24 h cultures of the non-toxigenic *S. epidermidis* strain did not induce proliferation, similar to unstimulated control cells. The lowest level of PBMC proliferation was observed in response to supernatants from cultures supplemented with 0.07% sorbic acid, both for coagulase-negative *S. epidermidis* 4S (6.67 ± 3.16, *p* = 0.007) and *S. aureus* FRI913 (18.07 ± 3.75, *p* = 0.0049).

## 3. Discussion

The enterotoxins produced by *S. aureus* are the cause of staphylococcal food poisoning. Within one to six hours after consuming food contaminated with SEs, patients typically present with symptoms of acute gastroenteritis, including severe vomiting and diarrhea [17]. However, these are not the only effects of consuming staphylococcal toxins. In addition to their emetic activity, SEs are known as mitogens, also called superantigens. The superantigenic properties of SEs are less understood in the context of food poisoning. Recent research indicates that the ingestion of enterotoxins with food may interfere with the gut immune system, contributing to the development of allergies [23], Crohn’s disease [24], and colonic inflammatory bowel disease [24,25]. Despite the lack of legal requirements for food testing for enterotoxins produced by coagulase-negative staphylococci (CNS), numerous scientific reports have raised concerns about the presence of toxin-producing strains in meat and meat products [4,26,27]. To date, the effect of food preservatives on the production of enterotoxins by coagulase-negative staphylococci (CNS) remains uninvestigated.

Generally, for *S. aureus*, preservatives that inhibit bacterial growth can also reduce enterotoxin production by limiting the number of bacteria present. However, some preservatives may stimulate enterotoxin production under certain conditions. *S. aureus* demonstrates remarkable physiological adaptability, maintaining proliferation and enterotoxin synthesis across diverse environmental parameters, including a broad range of temperatures, pH levels, water activity (aw), and sodium chloride concentrations [17,28,29]. Various regulatory mechanisms influence SE expression depending on the genetic element harboring the SE genes. These genes are located on diverse genetic elements, including plasmids, bacteriophages, and pathogenicity islands (SaPIs) [18,30].

Regassa et al. [31] showed that high concentrations of NaCl (1.2 M) inhibit production of SEC at the transcriptional level. The authors suggest high osmolarity, independently of the Agr system, as a reason for the reduced sec mRNA production. Osmoprotective compounds like choline, dimethylglycine, glycine betaine, and L-proline restore the ability to produce SEC. In another publication by the same author, it was demonstrated that the presence of glucose in the medium reduces the amount of extracellular SEC and sec gene transcripts. These studies also suggest that sec gene expression in the presence of glucose is not dependent on Agr [31]. This was also confirmed in studies by Etter et al. conducted on seven strains producing different SEC variants. Additionally, the authors demonstrated that Agr, SarA, and SigB do not influence the reduction in SEC production under glucose-induced stress [32]. It has been observed that expression of the *sec* gene at both the protein and mRNA levels proceeds independently. The authors also emphasize that the reduction in SEC production under glucose-induced stress does not apply to SECbov, as observed in two bovine isolates (both belonging to CC151 genotype and secreting SECbovine). This suggests that the regulation of SEC expression may be strain dependent. In our study, no significant changes in SEC protein levels, as measured by ELISA, were observed in the 2–4.5% NaCl, and in a mixture of 2% NaCl and 2% potassium nitrate in either of the tested strains. However, *S. aureus* produced significantly higher amounts of enterotoxin under these conditions compared to *S. epidermidis* 4S. At the transcriptional level, however, the presence of 4.5% NaCl may misleadingly suggest an inhibitory effect on sec gene expression in *S. epidermidis*.

Nitrites and nitrates are commonly used preservatives in meat production, known for their antimicrobial properties as well as their ability to enhance organoleptic qualities, such as the characteristic pink coloration of cured meat products [12,15,33,34,35]. In our study, potassium nitrate did not exhibit any inhibitory effect on the growth of staphylococci, including both coagulase-positive and coagulase-negative strains. The transcript level of the *sec* gene does not reflect protein production, as we observed a significant increase in sec mRNA levels for SEC_epi_ across all tested preservative concentrations, whereas no corresponding increase in SEC protein levels was detected. In the study of Etter et al. [33], investigating the effect of nitrite stress (150 mg/L) on SEC production in seven *S. aureus* strains, it was demonstrated that protein expression did not correlate with mRNA expression. The authors concluded that the use of nitrites as preservatives does not reduce SEC levels and may, in fact, increase SEC concentrations in certain strains, posing a potential risk to consumers. However, direct comparison with our results is challenging, as their study focused on SEC_1_, SEC_2_, SEC_bovine_, and SEC_ovine_ isoforms, while our research evaluates the effect of nitrates on SEC_3_ and SEC_epi_. Studies on SEC expression in response to food preservatives are limited, and existing comparisons of enterotoxins C, B, and D suggest distinct regulatory mechanisms governing their expression [32,33].

Our findings indicate that the only preservative that significantly inhibits the production of both SEC_epi_ and SEC_3_ at the mRNA and protein levels is 0.07%. sorbic acid. There are no reports on the effect of this compound on these enterotoxins. The effect of sorbic acid has been described for the expression of enterotoxin A [3]. The mechanism of regulation of SEA production is different from the SECs analyzed here; however, the effect of sorbic acid seems to be similar. During their experiments, the authors examined the expression and production of SEA by two *S. aureus* strains in the presence of NaCl, sorbic acid, and their mixtures. Incubation of *S. aureus* in 2% NaCl led to an increase in SEA production, which was attributed to the induction of the prophage carrying the sea gene. In contrast, sorbic acid inhibited SEA production and minimized the effect of osmotic shock after the addition of NaCl. The authors did not observe inhibition of *S. aureus* growth under the influence of sorbic acid [3]. Generally, such conditions as pH changes, low water activity, changes in osmotic balance, and other factors intended to inhibit bacterial growth enhance SEA expression [36].

*Staphylococcal* enterotoxins present in food are considered a cause of food poisoning when present at concentrations exceeding 10–20 ng/mL [29,37,38,39]. However, their mitogenic properties can be observed at concentrations below 1 ng [40]. An increasing number of reports suggest a potential role of staphylococcal enterotoxins in the development of immune-mediated intestinal diseases [24,25]. Therefore, culture supernatants obtained under the influence of the tested preservatives were also evaluated for their mitogenic properties. In samples where production of SEC_3_ and SEC_epi_ was significantly reduced by sorbic acid, we observed a markedly lower stimulation of PBMC proliferation compared to the control. Our previous studies in a mouse model demonstrated that SEC_3_ and SEC_epi_ contribute to severe morphological changes in the mouse intestinal epithelium and influence gut-associated lymphoid tissue (GALT) [19]. We also evaluated the effects of SEC_3_ and SEC_epi_ on activation and proliferation markers in T cells. In intraepithelial lymphocytes (IELs), expression of the activation marker CD69 on T cells began to increase as early as 4 h after exposure to SEC_epi_, whereas CD69 upregulation in response to *S. aureus* SEs (e.g., SEC_3_) was observed starting at 24 h post-treatment. There is an ongoing debate regarding the role of T cells in the gut. On one hand, they exhibit IL-10-mediated regulatory functions [40] that are protective for the gut, while on the other, they demonstrate direct cytotoxic activity that may contribute to the progression of inflammatory bowel disease [41]. Our findings underscore the need to evaluate food preservatives not only for antimicrobial efficacy but also for their impact on bacterial virulence.

Our study demonstrates the effect of food preservatives on the expression and production of SEC_3_ and SEC_epi_. While potassium nitrate and sodium chloride did not significantly inhibit bacterial growth or reduce enterotoxin levels, sorbic acid at a concentration of 0.07% decreased both mRNA expression and protein production of SEC_3_ and SEC_epi,_ and suppressed the mitogenic effect of *S. aureus* and *S. epidermidis* culture supernatants. These findings, in line with growing evidence of enterotoxin involvement in immune-mediated gut disorders, underscore the importance of evaluating preservative impact not only on bacterial viability but also on virulence factor expression and its immunosupresive effects. This is particularly relevant given the underestimated role of enterotoxins from coagulase-negative staphylococci and the potential health risks they may pose even at sub-emetic concentrations. 

This study has several limitations. First, the experiments were conducted using only three commonly used food preservatives—potassium nitrate, sodium chloride, and sorbic acid—which do not represent all categories of preservatives. Therefore, the results should be interpreted as preliminary observations rather than comprehensive conclusions. Second, the study focused primarily on the phenotypic effects of these preservatives on the production of Staphylococcal Enterotoxin C (SEC), without exploring the underlying molecular mechanisms involved in toxin regulation. Future studies including a wider range of preservative compounds and employing molecular and biochemical analyses will be necessary to clarify the mechanisms responsible for the observed effects.

## 4. Materials and Methods

### 4.1. Bacterial Strains and Growth Curve Determination

*S. epidermidis* 4S is derived from ready-to-eat food samples and is known as a strain harboring the gene enterotoxins C [22]. *S. epidermidis* 12228 and *S.aureur RN4220* ware donated by dr Keun Seok Seo as a nontoxigenic strain (negative controls). *S. aureus* FRI 913 strain was used in this study as a positive control, as it produces enterotoxin C. Bacterial strains were cultivated in a chemical-defined medium CY (yeast extract, tryptone) without any source of glucose (to exclude the influence of sugars on the expression of enterotoxins) with popular food preservatives: 2% NaCl; 2% KNO_3_; 4.5% NaCl; 4.5% KNO_3_; a mixture of 2% NaCl and 2% KNO_3_, 0.07% sorbic acid for 24 h at 37 °C with agitation (230 rpm). The concentration of 2% NaCl corresponds to typical levels found in processed meat products, as reported in recent food-matrix studies. The higher concentration of 4.5% NaCl was intentionally applied to simulate a stress condition that may occur in certain curing or salted meat surfaces under specific processing conditions. The concentration of 0.07% sodium nitrate was selected based on regulatory limits for cured meat products and on typical concentrations reported in industrial practice [42]. Growth curve OD_600_ was measured after 3, 5, 7 24, and 48 h. Using the dot blot method, after collecting 100 µL of post-culture fluid, the stage of the growth phase was selected for research on the effect on the expression of enterotoxins. Samples for the cell count, qPCR, and ELISA were collected at 24 h of culture. Bacterial cells were quantified by plating serial dilutions of the culture on BHI agar. All experiments were carried out in triplicate with three biological replicates.

### 4.2. Dot Blot

Semi-quantitative screening for protein expression was made using the dot blot technique. Post-culture fluids were collected after 3, 5, 7, 24 and 48 h after inoculation of the medium and medium with food preservatives with *S. aureus* and *S. epidermidis* 4S. 50 µL of fluid was poured on a PVDF membrane using a dot blotter (Roth, Karlsruhe, Germany), and incubated with rabbit anti–SEC antiserum (at 1:5000) (provided by dr Keun Seok Seo) for 1 h at RT. After three cycles of washing with PBST the membrane was incubated for 1 h in RT in anti-rabbit antibody conjugated with HRP (at 1:2000) (Santa Cruz, Heidelberg, Germany). After three cycles of washing with PBST membranes were incubated with a chemiluminescent substrate and exposed on X-ray film.

### 4.3. Sandwich ELISA

The concentrations of SECs in culture supernatants were evaluated using ELISA. For this, 1 mL of culture supernatants were centrifuged at 400× *g* for 10 min, and kept at −20 °C until measurement. Culture supernatants were preincubated with 20% normal rabbit serum overnight at 4 °C, to bind protein A, and diluted in PBS containing 0.01% Tween-20 for SEC-ELISA test. Because the production of SEC differed substantially among the tested strains and conditions, sample dilutions were adjusted individually (from 4× to 16×) to ensure that all measured values fell within the dynamic range of the standard curve (100–0.3 ng/mL). ELISA was performed according to the protocol described by Lis at al. [43]. Recombinant SEC [22], was used to determine the standard curve for the ELISA. Rabbit polyclonal anti-SEC antibody was purchased from OriGene Technologies, Inc. (Rockville, MD, USA). Antibodies were biotinylated with biotin N-hydroxysuccinimide ester (Merck, Darmstadt, Germany). A conjugate of HRP-streptavidin (Merck, Darmstadt, Germany) was used to detect biotinylated antibodies, and 3,3′,5,5′-tetramethylbenzidine (Merck, Darmstadt, Germany) was used as a substrate for HRP. Tested serum was diluted up to 75-times with PBS. The absorbance at 405 nm was read with a microplate reader (Spark Tecan, Männedorf, Switzerland). The concentration of the enterotoxins in the samples was measured using a 4-parameter logistic curve fit. All determinations were performed in triplicate. Data ware analyzed using GraphPad Prism software (version 9.5.0, GraphPad Software Inc., San Diego, CA, USA). Each experiment was carried out in four independent biological replicates, with three technical replicates for each biological sample.

### 4.4. RNA Extraction, Reverse Transcription, qPCR

Bacterial pellets from 1 mL of 24 h culture broth were suspended in Fenzol (A&A Bio-tech, Poland) and kept for 24 h et −20 for further analysis. RNA was isolated using an RNA isolation kit (A&A Biotech, Gdańsk, Poland) according to the manufacturer’s instructions, with previous mechanical cell wall disruption by adding 50 mg of glass beads, diameter 150–212 µm (Sigma-Aldrich, Taufkirchen, Germany) to each pellet and carried out in the Tissue Lyser LT (Qi-agen, Hilden, Germany). Three cycles of beating of 2 min each, with 1 min incubation on ice within cycles, were performed. RNA was dissolved in 50 μL of water and the concentration of RNA was measured using a spectrophotometer (DeNovix DS-11 FX, Wilmington, DE, USA) and quantified by measuring A260 and A280. 1 μg of RNA was treated with RNAse-free DNase I (Sigma-Aldrich, Taufkirchen, Germany) to eliminate residual genomic DNA. The integrity of RNA was evaluated by electrophoresis on 1% agarose gel. cDNA was synthesized using random hexamers and SuperScript III^®^ (Invitrogen, Carlsbad, CA, USA) following the manufacturer’s instructions. qPCR was carried out on a CFX Connect™ Real-Time System (Bio-Rad, Hercules, CA, USA), using SsoFast EvaGreen Supermix (Bio-Rad, Hercules, CA, USA). The reaction mixture contained 1 µL of template cDNA, 0.5 μM of each primer (listed in Table 1), 10 µL of SsoFast EvaGreen Supermix, and water up to 20 µL. Reaction mixtures were initially incubated for 30 s at 95 °C, followed by 35 cycles at 95 °C for 10 s and 15 s at 60 °C. The specificity of PCR was evaluated by melt curve analysis in a temperature range of 95–58 °C performed for each reaction. Residual DNA contamination was checked in each RNA sample by running no-RT controls. All reactions were performed in triplicate (technical replicates), and at least three independent biological replicates were analyzed. Data analysis was carried out using Bio-Rad CFX Manager (version 3.1, Bio-Rad, Hercules, CA, USA) software. The housekeeping gene used for normalization was selected based on its expression stability under exposure to food preservatives for *S. aureus*. It was chosen from eight candidate genes (*rpoB*, *16S rRNA*, *gyrB*, *recA*, *rho*, *pta*, *rplD*, and *tpo*), evaluated for stability using comparative analysis of expression variation across the tested condition. The BestKeeper software (https://www.gene-quantification.de/bestkeeper.html, accessed on 5 January 2022) was used as described previously [44]. The 16S rRNA gene showed the highest expression stability and was used as an internal reference for normalization. Primer specificity and efficiency were verified by melt curve analysis and standard curve determination, respectively; the efficiency of amplification for all primer pairs ranged from 95% to 105% with R^2^ ≥ 0.99. Relative gene expression levels of the enterotoxin gene were calculated using the 2^−ΔΔCt^ method, normalized to the 16S rRNA reference gene [45]. Data analysis was performed using Bio-Rad CFX Manager software (Bio-Rad, USA). Each experiment was carried out in four independent biological replicates, with three technical replicates for each biological sample

### 4.5. PBMC Proliferation Assay

Human PBMCs were obtained from healthy volunteers, who signed a written consent form, and were recruited into the study. The protocol was reviewed and approved by the Institutional Review Board at Mississippi State University (protocol code 13–191). Peripheral blood mononuclear cells (PBMCs) were isolated from three independent healthy donors, and each experiment was performed in technical triplicates. The results presented in the manuscript represent the mean values obtained from all donors. PBMC were obtained from the blood by centrifugation in a gradient using Histopaque-1077 (Sigma-Aldrich). The isolated PBMCs were washed twice in PBS and labeled with CFSE, using a CellTrace™ CFSE Cell Proliferation Kit (ThermoFisher Scientific, Waltham, MA, USA). Labeled cells were resuspended in RPMI-1640 medium supplemented with 10% heat-inactivated FBS, 2 mM L-glutamine, and 1% penicillin-streptomycin (Invitrogen, Carlsbad, CA, USA) at a final concentration of 5 × 10^5^ cells/mL. Labeled PBMC’s were incubated with 10-fold diluted post-cultured bacterial medium after the addition of food preservatives. Post-culture fluid from a culture of a non-toxic strain *S. epidermidis* 12228 and *S.aureus RN4220* was used as a negative control of lymphocyte proliferation stimulation. As a positive control, recombinant SEC at 1 ng/mL was used. The PBMCs were incubated in 96-well U-bottomed plates in 0.2 mL of culture medium, at 37 °C in a humidified atmosphere containing 5% CO_2_ for 6 days. Untreated, labeled PBMC’s kept in complete cell medium for the same time, served as a negative control. After incubation, cell division analysis was performed using a NovoCyte flow cytometer (Agilent Technologies, Santa Clara, CA, USA) and a Cytation 5 reader (Agilent BioTek, Santa Clara, CA, USA) and data were analyzed with FlowJo software (version 10.8.1, Tree Star, Salford, UK). Flow cytometric gating was applied to sequentially exclude debris (FSC/SSC), doublets (FSC-A vs. FSC-H), followed by selection of viable total lymphocytes for proliferation analysis.

### 4.6. Statistical Analysis

Statistical significance of the results was assessed using a one-way ANOVA test and the Mann–Whitney test. The statistical significance was considered with the following values: * *p* < 0.05, ** *p* < 0.01, *** *p* < 0.001. Statistical analyses were performed using Statistica version 12 (StatSoft Inc., Cracow, Poland) and GraphPad Prism 5.02 (GraphPad Software Inc., San Diego, CA, USA).

## Figures and Tables

**Figure 1 ijms-26-11659-f001:**
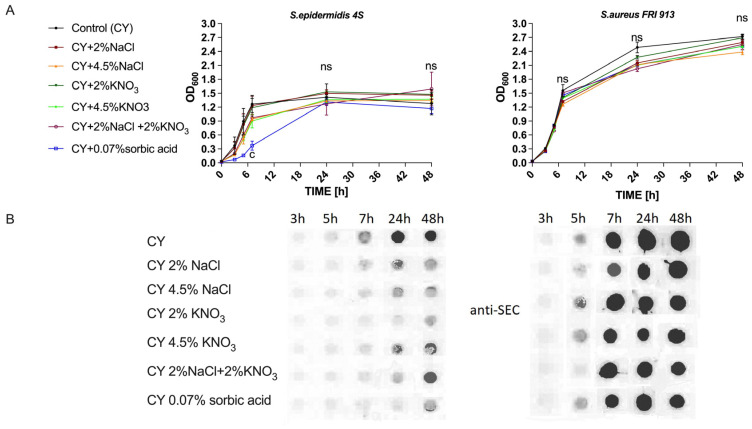
Growth curve of *Staphylococcus epidermidis* 4S and *S. aureus* FRI913 (**A**) and control dot blots assay (**B**) with anti-SEC polyclonal rabbit antibody with post-cultivation fluids in different conditions and time points. The means are presented from replicates of 3 independent experiments in 3 technical replicates.

**Figure 2 ijms-26-11659-f002:**
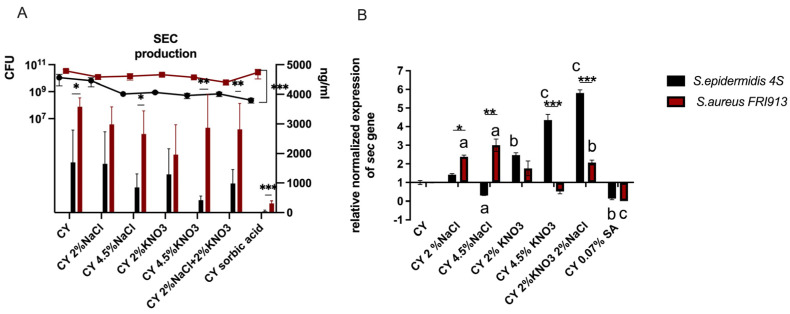
CFU and SEC production of *S. aureus* FRI913 (maroon square and bars consecutively) and *S. epidermidis* 4S (black square and bars consecutively) (**A**). Relative normalized expression. After 24 h, expression of *sec* gene (**B**) was measured by quantitative real-time PCR and compared to the expression level in medium without supplementation (first bar). The means are presented from replicates of 4 independent experiments in 3 technical replicates. Asterix indicates significant differences in the mean between strains, as determined by Mann–Whitney (* *p* < 0.05, ** *p* < 0.01, *** *p* < 0.001) and different letters indicate significant differences in the mean percentage between treatments, as determined by ANOVA, followed by the Tukey test (a *p* < 0.001, b *p* < 0.01, c *p* < 0.05).

**Figure 3 ijms-26-11659-f003:**
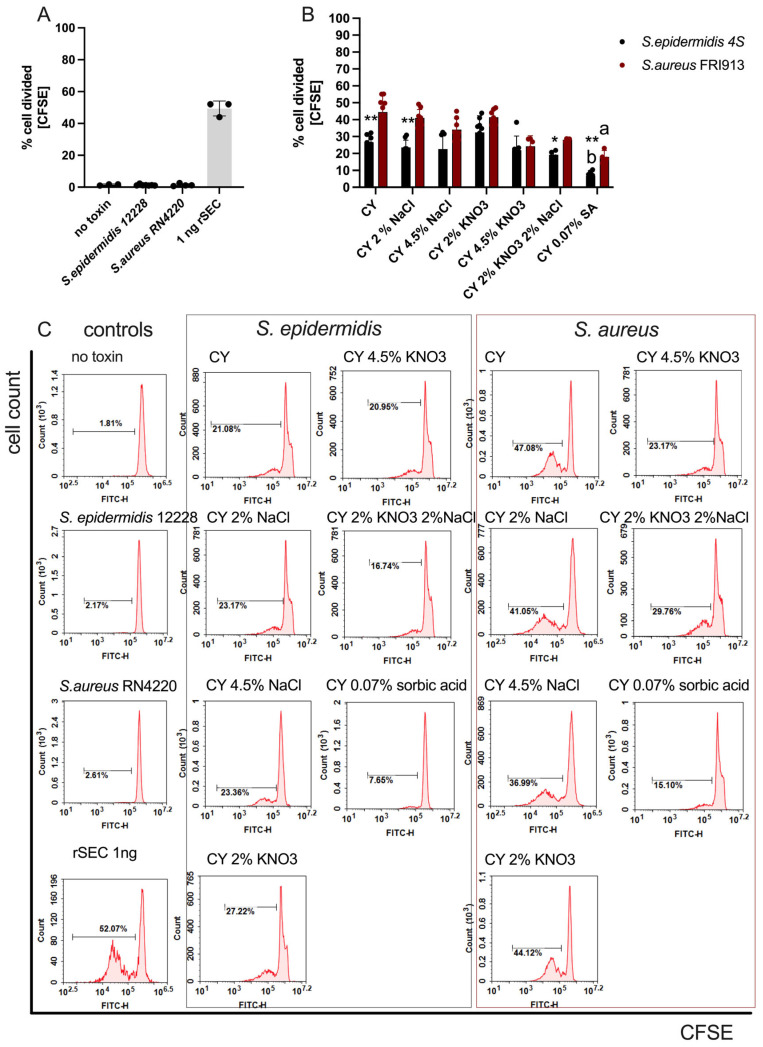
Percentage of proliferation stimulation of PBMCs in response to culture supernatants derived from *S. aureus* and *S. epidermidis* grown in the presence of various food preservatives (**A**), and (**B**) non-toxic *S. epidermidis* ATCC 12228 and *S.aureus 4220* (negative controls) and positive control rSEC 1 ng. Data represent three biological replicates; each performed in triplicate. Example histograms for each of the tested conditions (**C**). Asterix indicates significant differences in the mean between strains, as determined by Mann–Whitney (* *p* < 0.05, ** *p* < 0.01) and different letters indicate significant differences in the mean percentage between treatments, as determined by ANOVA, followed by the Tukey test (*p* < 0.001, *p* < 0.01, *p* < 0.05).

**Table 1 ijms-26-11659-t001:** Primer list.

Gene	Primers Sequence	Source
*sec*	F: 5′-CTC AAG AAC TAG ACA TAA AAG CTA GG-3′ R: 5′-TCA AAA TCG GAT TAA CAT TAT CC-3′	[26]
*16S*	F: 5′-AAGTCCCGCAACGAGCGCAA-3′ R: 5′-CCTCCGGTTTGTCACCGGCA-3′	[46]
*rpoB*	F:5′-CTACAAAACCAATTCCGTATCG-3′ R:5′-TTAATTGTTGAGGTGTGATAGAC-3′	[4]

## Data Availability

The data presented in this study are available on request from the corresponding author.

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
