# Peer review of "Do Food Preservatives Affect Staphylococcal Enterotoxin C Production Equally?"

_ijms, 2025, doi:10.3390/ijms262311659_

Round 1

Reviewer 1 Report

Comments and Suggestions for Authors

    This manuscript investigates the impact of commonly used food preservatives, including sodium chloride, potassium nitrate, and sorbic acid, on the production of Staphylococcal enterotoxins, particularly SEC and SECepi, in Staphylococcus aureus and Staphylococcus epidermidis. The authors employ ELISA, RT-qPCR, and PBMC proliferation assays to evaluate the expression and production of toxins, providing valuable insights into the potential for mitigating foodborne illness risks associated with these enterotoxins.

      In scientific writing, the genus and species names should always be italicized. In this manuscript, the species names Staphylococcus aureus and Staphylococcus epidermidis appear without italics, which is inconsistent with standard scientific conventions. I suggest correcting this throughout the manuscript for consistency and to adhere to proper formatting guidelines.

    In the introduction and discussion sections, additional literature on Staphylococcus aureus enterotoxins should be included, such as SEA, SEB. You can read and use these references:

  1. Liu, C. M., Liu, J. B., Wang, W. L., Yang, M., Chi, K. M., Xu, Y. Y., & Guo, N. (2023). Epigallocatechin Gallate Alleviates Staphylococcal Enterotoxin A-Induced Intestinal Barrier Damage by Regulating Gut Microbiota and Inhibiting the TLR4-NF-κB/MAPKs-NLRP3 Inflammatory Cascade. Journal of Agricultural and Food Chemistry, 71(43), 16286-16302. https://doi.org/10.1021/acs.jafc.3c04526.
  2. Chi, K. M., Zou, Y. P., Liu, C. M., Dong, Z. J., Liu, Y., & Guo, N. (2023). Staphylococcal enterotoxin A induces DNA damage in hepatocytes and liver tissues. Toxicon, 221. https://doi.org/10.1016/j.toxicon.2022.106980.
  3. Zhang, J., Wang, J., Jin, J., Li, X., Zhang, H. L., Shi, X. N., Zhao, C. (2022) Prevalence, antibiotic resistance, and enterotoxin genes of Staphylococcus aureusisolated from milk and dairy products worldwide: A systematic review and meta-analysis. Food Research International, 162. https://doi.org/10.1016/j.foodres.2022.111969.

    The manuscript reports exact p-values for various comparisons. While reporting p-values is important for statistical transparency, it is not always necessary to include the exact values, particularly when the focus is on statistical significance. I suggest considering whether it would be sufficient to report p-values as p<0.05, p<0.01, or p<0.001 to simplify the presentation and avoid redundancy, unless the exact values are essential for the context of the paper.

    In Figure 3C, the y-axis label appears unclear, which may hinder the reader's ability to fully interpret the data.

    It is important to maintain consistency in the use of units throughout the manuscript. For instance, the unit for time is written both as "hour" and "h" in different parts of the manuscript.

   Line 344, the expression of concentration appears to be unclear.

Comments on the Quality of English Language

The manuscript presents valuable findings, but its clarity and professionalism are currently limited by the quality of the English writing.

Author Response

Comment 1: In scientific writing, the genus and species names should always be italicized. In this manuscript, the species names Staphylococcus aureus and Staphylococcus epidermidis appear without italics, which is inconsistent with standard scientific conventions. I suggest correcting this throughout the manuscript for consistency and to adhere to proper formatting guidelines.

Response 1: Thank you for pointing this out. The bacterial species names have been revised to italic font throughout the manuscript.

Comment 2: In the introduction and discussion sections, additional literature on Staphylococcus aureus enterotoxins should be included, such as SEA, SEB. You can read and use these references:

  1. Liu, C. M., Liu, J. B., Wang, W. L., Yang, M., Chi, K. M., Xu, Y. Y., & Guo, N. (2023). Epigallocatechin Gallate Alleviates Staphylococcal Enterotoxin A-Induced Intestinal Barrier Damage by Regulating Gut Microbiota and Inhibiting the TLR4-NF-κB/MAPKs-NLRP3 Inflammatory Cascade. Journal of Agricultural and Food Chemistry, 71(43), 16286-16302. https://doi.org/10.1021/acs.jafc.3c04526.
  2. Chi, K. M., Zou, Y. P., Liu, C. M., Dong, Z. J., Liu, Y., & Guo, N. (2023). Staphylococcal enterotoxin A induces DNA damage in hepatocytes and liver tissues. Toxicon, 221. https://doi.org/10.1016/j.toxicon.2022.106980.
  3. Zhang, J., Wang, J., Jin, J., Li, X., Zhang, H. L., Shi, X. N., Zhao, C. (2022) Prevalence, antibiotic resistance, and enterotoxin genes of Staphylococcus aureusisolated from milk and dairy products worldwide: A systematic review and meta-analysis. Food Research International, 162. https://doi.org/10.1016/j.foodres.2022.111969.

Response 2: We sincerely thank the reviewer for the valuable suggestion to include additional literature on Staphylococcus aureus enterotoxins, particularly SEA and SEB. We fully recognize the importance of these classical emetic enterotoxins (SEA–SEE) in the broader context of S. aureus pathogenicity and foodborne illness. However, the scope of the present study is specifically focused on enterotoxin C and its comparison with the orthologous protein in S. epidermidis. The prevalence of the sea and seb genes among coagulase-negative staphylococci (CNS) is inconsistent in the literature (1, 2, 3). Some reports suggest a high prevalence of these genes; however, none of these studies provide evidence for the expression of enterotoxin B, and some even rule out its production despite positive PCR results. Based on long-term studies conducted in our laboratory, the presence of enterotoxin A in CNS from food in Poland is rare, and enterotoxin B is not observed. The most frequently detected enterotoxins among CNS are SED, SEH, and SEC; however, only enterotoxin C (SEC) has been fully characterized in terms of genomic location, stability, resistance to proteolytic digestion, and enterotoxigenic effect in in vivo studies (4,5,6).

Therefore, our study focuses specifically on SEC, as it is the only CNS enterotoxin with sufficient characterization to support a detailed comparative analysis with its ortholog in S. aureus (SEC3).

  1. Podkowik M, Park JY, Seo KS, Bystroń J, Bania J. Enterotoxigenic potential of coagulase-negative staphylococci. Int J Food Microbiol. 2013 Apr 15;163(1):34-40. doi: 10.1016/j.ijfoodmicro.2013.02.005. Epub 2013 Feb 16. PMID: 23500613; PMCID: PMC6671284.
  2. Zell, C., Resch, M., Rosenstein, R., Albrecht, T., Hertel, C., Götz, F., 2008. Characterization of toxin production of coagulase-negative staphylococci isolated from food and starter cultures. International Journal of Food Microbiology 127, 246–251
  3. Moura TM, Campos FS, d'Azevedo PA, Van Der Sand ST, Franco AC, Frazzon J, Frazzon AP. Prevalence of enterotoxin-encoding genes and antimicrobial resistance in coagulase-negative and coagulase-positive Staphylococcus isolates from black pudding. Rev Soc Bras Med Trop. 2012 Oct;45(5):579-85. doi: 10.1590/s0037-86822012000500008. PMID: 23152340.
  4. Podkowik M, Seo KS, Schubert J, Tolo I, Robinson DA, Bania J, Bystroń J. Genotype and enterotoxigenicity of Staphylococcus epidermidis isolate from ready to eat meat products. Int J Food Microbiol. 2016 Jul 16;229:52-59. doi: 10.1016/j.ijfoodmicro.2016.04.013. Epub 2016 Apr 14. PMID: 27105039; PMCID: PMC4877272.
  5. Gonet M, Krowarsch D, Schubert J, Tabiś A, Bania J. Stability and Resistance to Proteolysis of Enterotoxins SEC and SEL Produced by Staphylococcus epidermidisand Staphylococcus aureus. Foodborne Pathog Dis. 2023 Jan;20(1):32-37. doi: 10.1089/fpd.2022.0059. PMID: 36622956.
  6. Tabiś A, Gonet M, Schubert J, Miazek A, Nowak M, Tomaszek A, Bania J. Analysis of enterotoxigenic effect of Staphylococcus aureus and Staphylococcus epidermidis enterotoxins C and L on mice. Microbiol Res. 2022 May;258:126979. doi: 10.1016/j.micres.2022.126979. Epub 2022 Feb 4. PMID: 35158299.

In response to the reviewer’s suggestion, a sentence explaining the rationale for focusing on enterotoxin C has been added (line 65).

Comment 3:  The manuscript reports exact p-values for various comparisons. While reporting p-values is important for statistical transparency, it is not always necessary to include the exact values, particularly when the focus is on statistical significance. I suggest considering whether it would be sufficient to report p-values as p<0.05, p<0.01, or p<0.001 to simplify the presentation and avoid redundancy, unless the exact values are essential for the context of the paper.

Response 3: We appreciate the reviewer’s thoughtful comment and suggestion regarding the reporting of p-values. We agree that, in many cases, summarizing statistical significance using thresholds (e.g., p < 0.05, p < 0.01, p < 0.001) can simplify the presentation. However, we chose to report exact p-values throughout the manuscript to enhance statistical transparency and allow readers to better assess the strength of the evidence. This approach follows current recommendations from major journals and reporting guidelines (e.g., APA, Nature, and Elsevier), which encourage reporting exact p-values whenever available. We believe that providing these values contributes to clarity and reproducibility of our results. Nevertheless, we have carefully reviewed the manuscript and ensured that p-values are reported consistently and only where they add meaningful information.

Comment 4: In Figure 3C, the y-axis label appears unclear, which may hinder the reader's ability to fully interpret the data.

Response 4: The quality of the figure has been improved.

Comment 5 :It is important to maintain consistency in the use of units throughout the manuscript. For instance, the unit for time is written both as "hour" and "h" in different parts of the manuscript.

Response 5: has been unified

Comment 6:  Line 344, the expression of concentration appears to be unclear.

Response 6: has been corrected to: of 5 × 105 cells/ml.

Comment 7: The manuscript presents valuable findings, but its clarity and professionalism are currently limited by the quality of the English writing.

Response 7: English proofreading was carried out by a native speaker

Reviewer 2 Report

Comments and Suggestions for Authors

This study requires further clarification on why only SEC is analyzed, rather than other enterotoxins such as SEA and SEB.
If the goal is to investigate the effects of different food preservatives on enterotoxin production, the tested preservatives in the study are clearly insufficient, and the specific types of tested food preservatives (e.g., inorganic salts, organic acids) are not explicitly stated.
Why is sodium nitrate tested for its impact on staphylococcal toxin production? Is sodium nitrate indeed a food preservative?
Microbial names in the text are not italicized; the abbreviation "CNS" is used without its full form being explained upon first occurrence.
In most figures, the significance markers (e.g., asterisks) are blurry or not aligned at the same height, requiring the figures to be recreated. Specifically, Figure 3c is entirely illegible.
Line 223 contains an extra period.

Author Response

Comment 1: This study requires further clarification on why only SEC is analyzed, rather than other enterotoxins such as SEA and SEB:

Answer 1: We sincerely thank the reviewer for the valuable suggestion to include additional reaserch on Staphylococcus aureus enterotoxins, particularly SEA and SEB. We fully recognize the importance of these classical emetic enterotoxins (SEA–SEE) in the broader context of S. aureus pathogenicity and foodborne illness. However the present study focuses specifically on enterotoxin C (SEC) and its comparison with the orthologous enterotoxin identified in S. epidermidis following exposure to preservatives commonly used in meat production.The prevalence of the sea and seb genes among coagulase-negative staphylococci (CNS) is inconsistent in the literaturę (1, 2, 3). Some reports suggest a high prevalence of these genes; however, none of these studies provide evidence for the expression of enterotoxin B, and some even rule out its production despite positive PCR results. Based on long-term studies conducted in our laboratory, the presence of enterotoxin A in CNS from food in Poland is rare, and enterotoxin B is not observed. The most frequently detected enterotoxins among CNS are SED, SEH, and SEC; however, only enterotoxin C (SEC) has been fully characterized in terms of genomic location, stability, resistance to proteolytic digestion, and enterotoxigenic effect in in vivo studies (4,5,6).

Therefore, our study focuses specifically on SEC, as it is the only CNS enterotoxin with sufficient characterization to support a detailed comparative analysis with its ortholog in S. aureus (SEC3).

  1. Podkowik M, Park JY, Seo KS, Bystroń J, Bania J. Enterotoxigenic potential of coagulase-negative staphylococci. Int J Food Microbiol. 2013 Apr 15;163(1):34-40. doi: 10.1016/j.ijfoodmicro.2013.02.005. Epub 2013 Feb 16. PMID: 23500613; PMCID: PMC6671284.
  2. Zell, C., Resch, M., Rosenstein, R., Albrecht, T., Hertel, C., Götz, F., 2008. Characterization of toxin production of coagulase-negative staphylococci isolated from food and starter cultures. International Journal of Food Microbiology 127, 246–251
  3. Moura TM, Campos FS, d'Azevedo PA, Van Der Sand ST, Franco AC, Frazzon J, Frazzon AP. Prevalence of enterotoxin-encoding genes and antimicrobial resistance in coagulase-negative and coagulase-positive Staphylococcus isolates from black pudding. Rev Soc Bras Med Trop. 2012 Oct;45(5):579-85. doi: 10.1590/s0037-86822012000500008. PMID: 23152340.
  4. Podkowik M, Seo KS, Schubert J, Tolo I, Robinson DA, Bania J, Bystroń J. Genotype and enterotoxigenicity of Staphylococcus epidermidis isolate from ready to eat meat products. Int J Food Microbiol. 2016 Jul 16;229:52-59. doi: 10.1016/j.ijfoodmicro.2016.04.013. Epub 2016 Apr 14. PMID: 27105039; PMCID: PMC4877272.
  5. Gonet M, Krowarsch D, Schubert J, Tabiś A, Bania J. Stability and Resistance to Proteolysis of Enterotoxins SEC and SEL Produced by Staphylococcus epidermidisand Staphylococcus aureus. Foodborne Pathog Dis. 2023 Jan;20(1):32-37. doi: 10.1089/fpd.2022.0059. PMID: 36622956.
  6. Tabiś A, Gonet M, Schubert J, Miazek A, Nowak M, Tomaszek A, Bania J. Analysis of enterotoxigenic effect of Staphylococcus aureus and Staphylococcus epidermidis enterotoxins C and L on mice. Microbiol Res. 2022 May;258:126979. doi: 10.1016/j.micres.2022.126979. Epub 2022 Feb 4. PMID: 35158299.

Comment 2: If the goal is to investigate the effects of different food preservatives on enterotoxin production, the tested preservatives in the study are clearly insufficient, and the specific types of tested food preservatives (e.g., inorganic salts, organic acids) are not explicitly stated.

Answer 2: We thank the reviewer for this valuable comment. We agree that a broader range of food preservatives could provide a more comprehensive understanding of their effects on enterotoxin production. However, the goal of the present study was not to evaluate all possible preservatives, but rather to investigate the response of Staphylococcus aureus and S. epidermidis to selected compounds most commonly used in meat processing.

Specifically, we focused on inorganic salts (sodium chloride and sodium nitrate and sorbic acid), as these substances are widely applied in cured and processed meat products. The selection was guided by their practical relevance to real food systems and by the need to ensure that the tested preservatives did not completely inhibit bacterial growth. This approach was crucial, since inhibition of bacterial proliferation would suppress enterotoxin production through a different mechanism, making it impossible to distinguish between growth inhibition and specific regulatory effects on toxin synthesis.

This clarification has been added to the revised version of the manuscript.

The effect of organic salts on enterotoxin production is currently under investigation and will be presented soon in a separate manuscript.

Comment 3: Why is sodium nitrate tested for its impact on staphylococcal toxin production? Is sodium nitrate indeed a food preservative?

Answer 3: Sodium nitrate (NaNO₃) is widely used in cured meats such as ham, bacon, and sausages.  Helps meat retain a pink-red color by forming nitrosomyoglobin. Sodium nitrate itself is not strongly antibacterial, but it can be reduced by bacteria to sodium nitrite (NaNO₂), which is a potent inhibitor of Clostridium botulinum and other bacteria. Sodium nitrate is tested for its impact on staphylococcal toxin production because it is commonly used in meat preservation, but its effects on bacterial metabolism and toxin expression are not fully straightforward

Comment 4: Microbial names in the text are not italicized; the abbreviation "CNS" is used without its full form being explained upon first occurrence.

Answer 4: Thank you for pointing this out. The bacterial species names have been revised to italic font throughout the manuscript

Comment 5: In most figures, the significance markers (e.g., asterisks) are blurry or not aligned at the same height, requiring the figures to be recreated. Specifically, Figure 3c is entirely illegible.

Answer 5: The quality of the figure has been improved.

Comment 6: Line 223 contains an extra period.

Answer 6: Has been improved.

Reviewer 3 Report

Comments and Suggestions for Authors

This manuscript investigates the impact of three common food preservatives—sodium chloride, potassium nitrate, and sorbic acid—on the expression and production of Staphylococcal enterotoxin C (SEC3 in S. aureus and SECepi in S. epidermidis). The study integrates ELISA, RT-qPCR, and PBMC proliferation assays to connect toxin levels with functional immunological effects.

Major points:
The rationale for chosen concentrations (2%, 4.5%, 0.07%) should be better justified with reference to in situ food matrix levels or regulatory limits. For instance, 4.5% NaCl is quite high and not reflective of typical curing concentrations.

  • Optical density alone is insufficient to assert “no growth inhibition.” CFU counts at multiple timepoints would strengthen this conclusion.
  • It is unclear how many biological replicateswere used in each ELISA and qPCR test. “Four independent experiments with three technical replicates” is acceptable, but this must be explicitly stated under each figure legend.
  • A non-toxigenic  aureusstrain should be included alongside S. epidermidis 12228 to rule out species-specific background effects.
  • Informations according the MIQE guidelines is missed.
  • Details of standard curve range and limit of detection should be provided. The values (e.g., 1850 ng/mL vs. 6.8 ng/mL) differ by orders of magnitude—some normalization or log-scaling might clarify comparisons.
  • The manuscript mentions “PBMCs from healthy volunteers” but does not specify how many donors were used or whether each experiment was repeated with multiple donors.
  • No information is provided on flow cytometric gating, which is essential for assessing proliferation accurately. Key missing details:
    • How doublets and dead cells were excluded
    • Which cell subset was analyzed (total lymphocytes, CD3⁺, CD4⁺, or CD8⁺ T cells?)

Minor points:
quantitative data in the abstract is missed to strengten the interest for readers.

  • Figure 1 Caption: B is missed. Also the y-axis needs a unit, also it is dimensionless. Please add [-] on the y-axis. Also the organism names need to be written in italic in figure 1.
  • While the discussion is okay, the discussion occasionally overextends into unrelated immunological territory (e.g., double-positive T cells) that is not directly measured here. Condensing that section would improve focus.
  • Figure 3 C is not readable and very blurry.
  • The information about electrophoretic analysis of RNA is incomplete. Buffer, pretreatment of RNA etc. are missed.

Author Response

Comment: The rationale for chosen concentrations (2%, 4.5%, 0.07%) should be better justified with reference to in situ food matrix levels or regulatory limits. For instance, 4.5% NaCl is quite high and not reflective of typical curing concentrations.

Answer: We thank the reviewer for this insightful comment. In our revised manuscript, we have clarified the rationale for the selected concentrations. The concentration of 2% NaCl corresponds to typical levels found in processed meat products, as reported in recent food-matrix studies. The higher concentration of 4.5% NaCl was intentionally applied to simulate a stress condition that may occur in certain curing or salted meat surfaces under specific processing conditions. Although 4.5% NaCl may appear relatively high, some cured meat products can contain salt levels reaching up to 7%, as described in the literature.

The concentration of 0.07% sodium nitrate was selected based on regulatory limits for cured meat products and on typical concentrations reported in industrial practice and the literature .

These clarifications have been added to the revised manuscript to highlight the experimental and practical relevance of the selected preservative concentrations.

Chen Q, Cao J, She W, et al. Salt reduction in cured meat products: a review on strategies and mechanisms. Food Science and Human Wellness, 2025, 14(3): 9250056. https://doi.org/10.26599/FSHW.2024.9250056

Comment: Optical density alone is insufficient to assert “no growth inhibition.” CFU counts at multiple timepoints would strengthen this conclusion.

Answer: The authors are aware that optical density measurements provide only a general indication of growth trends and do not directly reflect viable cell counts. Our intention, however, was to determine growth curves under the tested conditions to ensure that the applied preservative concentrations did not markedly affect bacterial proliferation and, consequently, did not bias enterotoxin production. Figure 1, on the other hand, was designed to estimate the optimal time point for enterotoxin production rather than to provide detailed growth kinetics.

In our study, CFU counts were determined for the same time point at which enterotoxin concentrations were measured (Figure 2). Concusion “no growth inhibition” is based on CFU after 24h of inoculation.

Comment: It is unclear how many biological replicateswere used in each ELISA and qPCR test. “Four independent experiments with three technical replicates” is acceptable, but this must be explicitly stated under each figure legend.

Answer: has been added to the figure description

Comment: A non-toxigenic  S. aureus strain should be included alongside S. epidermidis 12228 to rule out species-specific background effects.

Answer: Has been improved.

Comment: Informations according the MIQE guidelines is missed:

Answer: These clarifications have been added to the revised manuscript:

All reactions were performed in triplicate (technical replicates), and at least three independent biological replicates were analyzed. Data analysis was carried out using Bio-Rad CFX Manager (Bio-Rad, USA) software. The housekeeping gene used for normalization was selected based on its expression stability under exposure to food preservatives for S. aureus. It was chosen from eight candidate genes (rpoB, 16S rRNA, gyrB, recA, rho, pta, rplD, and tpo), evaluated for stability using comparative analysis of expression variation across tested condition. The BestKeeper software (https://www.gene-quantification.de/bestkeeper.html, accessed on 5 January 2022) was used as described previously [43]. The 16S rRNA gene showed the highest expression stability and was used as an internal reference for normalization. Primer specificity and efficiency were verified by melt curve analysis and standard curve determination, respectively; the efficiency of amplification for all primer pairs ranged from 95% to 105% with R² ≥ 0.99. Relative gene expression levels of the enterotoxin gene were calculated using the 2^–ΔΔCt method, normalized to the 16S rRNA reference gene [44]

Comment: Details of standard curve range and limit of detection should be provided. The values (e.g., 1850 ng/mL vs. 6.8 ng/mL) differ by orders of magnitude—some normalization or log-scaling might clarify comparisons.

Answer: We thank the reviewer for this important comment. The standard curve for SEC quantification covered a concentration range from 100 ng/mL to 0.39 ng/mL, with an R² value greater than 0.99 and amplification efficiency within the acceptable range. The limit of detection (LOD) for the assay was approximately 0.3 ng/mL.

Because the production of SEC differed substantially among the tested strains and conditions, sample dilutions were adjusted (from 4× to 16×) to ensure that all measured values fell within the dynamic range of the standard curve. This approach was necessary to maintain measurement accuracy and avoid signal saturation.

Comment: The manuscript mentions “PBMCs from healthy volunteers” but does not specify how many donors were used or whether each experiment was repeated with multiple donors

Answer: We thank the reviewer for this valuable comment. Peripheral blood mononuclear cells (PBMCs) were isolated from three independent healthy donors, and each experiment was performed in technical triplicates. The results presented in the manuscript represent the mean values obtained from all donors. This information has now been added to the revised version of the Materials and Methods section.

Comment: No information is provided on flow cytometric gating, which is essential for assessing proliferation accurately. Key missing details:

    • How doublets and dead cells were excluded
    • Which cell subset was analyzed (total lymphocytes, CD3⁺, CD4⁺, or CD8⁺ T cells?)

Answer: Details on the flow cytometric gating strategy and the analyzed cell population have been added in the Methods section:

„Flow cytometric gating was applied to sequentially exclude debris (FSC/SSC), doublets (FSC-A vs. FSC-H), followed by selection of viable total lymphocytes for proliferation analysis.”

Minor points:
quantitative data in the abstract is missed to strengten the interest for readers.

  • Figure 1 Caption: B is missed. Also the y-axis needs a unit, also it is dimensionless. Please add [-] on the y-axis. Also the organism names need to be written in italic in figure 1.
  • While the discussion is okay, the discussion occasionally overextends into unrelated immunological territory (e.g., double-positive T cells) that is not directly measured here. Condensing that section would improve focus.
  • Figure 3 C is not readable and very blurry.
  • The information about electrophoretic analysis of RNA is incomplete. Buffer, pretreatment of RNA etc. are missed.

Has been improved

Round 2

Reviewer 3 Report

Comments and Suggestions for Authors

The manuscript has improved significantly—well done!

There are still a few minor issues to address. For example, some organism names are not italicized (e.g., line 19, Figure 3A). Additionally, Staphylococcus is written out in full when first introduced in line 32, but the full name is still used again in line 43 instead of the abbreviated form.

One more question: if the calibration range is 0.39 to 100, how can values above this range—such as 1850—be considered reliable?

Author Response

Comment1: There are still a few minor issues to address. For example, some organism names are not italicized (e.g., line 19, Figure 3A). Additionally, Staphylococcus is written out in full when first introduced in line 32, but the full name is still used again in line 43 instead of the abbreviated form.

Answer 1: Thank you for raising this important point. Has been improved.

comment 2 : One more question: if the calibration range is 0.39 to 100, how can values above this range—such as 1850—be considered reliable?

Answer 2: The majority of commercially and non- commercially available ELISA assays have a limited dynamic range, and in our case, linearity and precision are guaranteed only between 0.39 and 100. To ensure the reliability and accuracy of results that naturally fall above this range, our laboratory implements the standard and well-accepted practice of sample dilution.

Specifically, our standard operating procedure for all ELISA tests involves the application of multiple, varying dilution factors. This approach is crucial, as it ensures that the resulting optical density (OD) readings from the diluted samples consistently fall within the established, linear portion of the standard curve, allowing for reliable interpolation rather than unreliable extrapolation. We wish to confirm that this standard laboratory practice, including the use of multiple dilutions to bring high-concentration samples within the calibrated range, is detailed in the Materials and Methods section of the manuscript (line 320).